# Bright electron bunches from a plasma-wakefield accelerator with a steep density down-ramp

J. C. Wood ®[1] ✉, L. Boulton[1,2], J. Beinortaitė ®[1,3], J. Björklund Svensson ®[1], S. Bohlen ®[1], G. Boyle ®[1], J. M. Garland[1], P. Gonzalez Caminal[1], C. A. Lindstrøm ®[1], G. Loisch ®[1], S. M. Mewes ®[1], T. Parikh[1], F. Peña ®[1,4], K. Põder ®[1], S. Schröder ®[1], M. Thévenet ®[1], S. Wesch ®[1], J. Osterhoff ®[1] & R. D'Arcy[1]

High-brightness electron bunches drive fundamental research in particle physics and photon science. Key to achieving a high brightness is to have a low transverse emittance, which ensures that the bunch can be tightly focussed. In radiofrequency accelerators a low initial emittance can be rapidly degraded due to space charge forces, which are greatly diminished once the electron bunch attains a relativistic velocity. A plasma accelerator can maintain orders-of-magnitude higher accelerating fields than radiofrequency accelerators, while multiple techniques exist to create a low emittance electron bunch directly inside the plasma accelerator structure. Plasma accelerators therefore offer a possibility to create high-brightness bunches in wakefields driven even by low-quality drive bunches. Here we demonstrate the injection and gigavolt-per-metre acceleration of electron bunches with mm·mrad normalised emittance, $\mathcal{O}(10\,\mathrm{pC/MeV})$ spectral density and per-cent-level energy spread, all with excellent reproducibility.

Radiofrequency particle accelerators based on metallic cavities drive fundamental research in particle physics and photon science, but the rate at which they can accelerate particles is limited to -100 MeV m⁻¹ by the onset of electrical breakdown inside the accelerating structure. As a result, the accelerators for free-electron lasers (FELs) are several hundreds of metres long, and particle colliders reach tens of kilometres in length. An emerging and significantly more compact alternative is the plasma-based accelerator (PBA), which has no such limit as it is already made of broken-down material. In a PBA, an intense laser pulse or charged particle beam driver propagates through a plasma and expels electrons from its vicinity via its ponderomotive force or space-charge field, respectively[1,2]. The charge density wave left in its wake, the plasma wakefield, is used to rapidly accelerate a trailing bunch at rates of order (1–100 GV m⁻¹) in most experiments to date. Major results in beam-driven plasma acceleration (termed plasma wakefield acceleration or PWFA) include multi-GeV energy gains with

few-per-cent energy spreads[3,4] and energy-spread preservation with >40% energy-transfer-efficiency[5]. The energy doubling of high-energy electrons has also been demonstrated, where the rear of a bunch was accelerated in the wakefield driven by the front of the bunch[6]. The wakefield has linear transverse focusing fields, enabling the rapid acceleration of an electron bunch without increasing its normalised emittance $\epsilon_n$[7]. Low $\epsilon_n$ bunches are desirable since the property is inversely proportional to the 6D beam brightness $B_{6D} = I/(\epsilon_{n,x}\epsilon_{n,y}\sigma_E)$ where $I$ is the beam current and $\sigma_E$ the energy spread. A high $B_{6D}$ is required to drive brilliant FELs, while a large $B_{4D} = Q/(\epsilon_{n,x}\epsilon_{n,y})$, where $Q$ is the beam charge, is a key ingredient in the luminosity of colliders.

The large accelerating field of a plasma wakefield can also be exploited to create a plasma injector, in which electrons are rapidly accelerated from rest to relativistic energies. This greatly reduces the effects of space-charge-driven emittance growth, which is a major limiting factor for the injectors for high-current RF accelerators. A

[1]Deutsches Elektronen-Synchrotron DESY, Hamburg, Germany. [2]SUPA, Department of Physics, University of Strathclyde, Glasgow, UK. [3]University College London, London, UK. [4]Institute of Experimental Physics, Universität Hamburg, Hamburg, Germany. ✉e-mail: jonathan.wood@desy.de

PWFA could, therefore, provide lower emittance electron bunches than is possible with radiofrequency injectors. The generation of high brightness bunches in plasma accelerators is also central to schemes to drive compact hard-x-ray FELs with saturation lengths of order 10 m[8]. Several schemes have been proposed to inject ultra-low-emittance, short-pulse, high-brightness electron bunches into a PWFA, either from ionisation processes within the wakefield[9–14], or via density-downramp injection (DDRI)[15–24]. In both sets of schemes above, there are advantages to electron beam rather than laser drivers, since radiofrequency electron linacs are already capable of accelerating tens of thousands of bunches per second with high efficiency[25], and extremely intense laser drivers preclude the possibility of controlled ionisation within the wakefield[13] or the production of truly customisable plasma density profiles.

Here, we focus on the DDRI scheme. In DDRI, a downramp is created in the longitudinal plasma density profile. Downramps can be generated simply and flexibly in a PWFA, for example, by using a moderate-intensity laser pulse orthogonal to the driver bunch to ionise additional gas species[22,23,26]. Since the wake wavelength increases as the density $n_e$ decreases, similar to the plasma wavelength $\lambda_p \propto 1/\sqrt{n_e}$, the rear of the wakefield is found progressively further behind the driver as a driver bunch travels through a downramp. Therefore its velocity drops significantly below that of the driver ($\sim c$), allowing background plasma electrons with high longitudinal velocities to be trapped in the wakefield and subsequently accelerated. Although the injected electrons initially receive a large transverse impulse from the drive beam, as they approach the back of the wake they experience a magnetic field from the radial current of the plasma electrons that form the wake structure. This generates a defocussing force, significantly reducing the transverse momentum and emittance of the injected electrons[27,28]. It has been shown in simulations that DDRI can produce bunches with $\epsilon_n < 100$ nm[21,27–29], which is comparable to the very best conventional FEL driver linacs today[30].

To date, experimental work on injection in PWFAs (e.g. by DDRI) has either shown high spectral density ($dQ/dE$) bunches with relatively high $\epsilon_n$ ~10 mm·mrad[23], or few-mm-mrad emittance bunches with $dQ/dE < 1$ pC MeV$^{-1}$[14]. In this work, we experimentally demonstrate the injection into a PWFA of 1.2 mm·mrad normalised emittance, 1.3% energy spread ($\sigma_E$) bunches with 14 pC MeV$^{-1}$ spectral density. The average 3D brightness $B_{3D} = \langle dQ/dE \rangle / \epsilon_{n,x}$ was 11.3 pC/MeV/mm·mrad. This represents a significant step forwards in the quality of plasma-generated electron bunches. We achieved this using the DDRI mechanism with an especially high-density injection region with steep density gradients, far beyond what has been considered previously. We describe the physics of this scheme with particle-in-cell (PIC) simulations. Full 3D PIC simulations recreating the experimental conditions as faithfully as possible suggest the normalised emittance of the injected beam in our setup could become as low as $0.14 \times 0.55$ mm·mrad.

## Results

### Experimental description

The experiment, depicted in Fig. 1a, was performed at the FLASHForward facility[31]. A ~50 mm long plasma, consisting mostly of Ar$^+$ and Ar neutrals, was created by the long-focal-length longitudinal laser pulse (further laser and plasma details can be found in the Methods section). A transversely propagating laser pulse was tightly focused through a narrow entrance hole at $z_h = 20$ mm after the capillary entrance, which ionised a small region of plasma up to Ar$^{4+}$. As a result, the on-axis plasma density profile shown in Fig. 1c exhibited a large density spike, the edge of which was used for DDRI[22,23,26]. The longitudinal laser underwent significant ionisation defocusing while producing the plasma[32], leading to the decaying density as a function of propagation distance. Only a small fraction of atoms were ionised for much of the accelerator length, although $n_e(z > z_h) \approx 2 \times 10^{16}$ cm$^{-3}$ was sufficiently

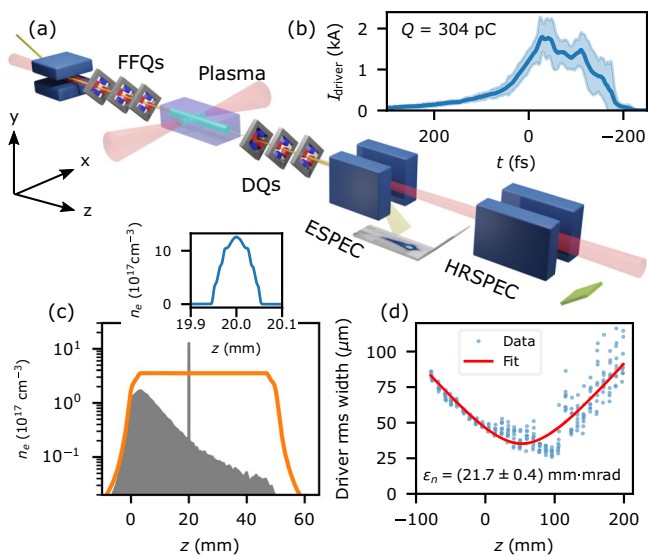

**Fig. 1 | Experimental setup. a** The longitudinal and transverse laser pulses (red) created the plasma (cyan). The driver bunch (yellow) propagated collinearly with the longitudinal laser from left to right. It was focused into the plasma cell by the final-focusing quadrupoles (FFQs). Electron bunches were imaged by the downstream quadrupoles (DQs) on to the electron spectrometer (ESPEC) screen (a Lanex screen placed after a 1 mm steel chamber wall), or the high-resolution ESPEC (HRSPEC) screen (an in-vacuum GAGG:Ce scintillator), after dispersion by the associated dipole magnet (blue). **b** Current profile of the driver bunch measured by a downstream transverse deflection structure. The shaded area represents the standard deviation. **c** Simulated longitudinal gas density (orange) and ionised plasma density (grey shaded area). Inset: simulated plasma density profile produced by the transverse laser. **d** Non-interacted driver bunch widths from an object-plane scan on the HRSPEC.

high to accelerate injected bunches to relativistic energies. The transverse laser pulse did not undergo significant ionisation defocussing.

The driving electron bunch was produced by the FLASH linac[33]. Its charge was $(304 \pm 2)$ pC and its mean energy was 689 MeV, with a root-mean-square energy spread of 5 MeV and a peak spectral density $dQ/dE = 47$ pC MeV$^{-1}$. Its current profile was measured using an X-band transverse deflecting structure (TDS) downstream of the setup[34,35]. The current profile that maximised injected charge is shown in Fig. 1b with a peak current $I_{pk} = (1.9 \pm 0.2)$ kA and a $(96 \pm 6)$ fs rms duration. All uncertainties here are the standard deviation. It was focused in to the plasma using the final-focusing quadrupoles (FFQs) to drive the plasma wakefield. Both drive and injected bunches were characterised with one of two imaging spectrometers: a broadband electron spectrometer (ESPEC) used for energy and divergence measurements, and a high-resolution spectrometer (HRSPEC) used for emittance measurements. The imaging was performed using a quadrupole triplet downstream of the capillary as shown in Fig. 1a. An example object-plane ($z_{obj}$) scan of the non-interacted driver performed with the downstream quadrupoles on the HRSPEC, done to measure its emittance, is shown in Fig. 1d (see 'Methods'). The normalised emittance $\epsilon_{n,x} = \frac{1}{m_e c}\sqrt{\langle x^2 \rangle \langle p_x^2 \rangle - \langle x p_x \rangle^2}$, where $x$ and $p_x$ are the position and momentum coordinates of bunch electrons, of a typical, non-interacted driver bunch was $\epsilon_{n,x} = (20 \pm 2)$ mm·mrad. $\epsilon_{n,y}$ could not be measured because this was the dipole dispersion direction.

### Optimisation of the injected bunch

After performing the spatio-temporal overlap procedure detailed in the 'Methods' section, the injected bunch was first characterised on the ESPEC with the imaging energy set to 30 MeV. Electrons were only

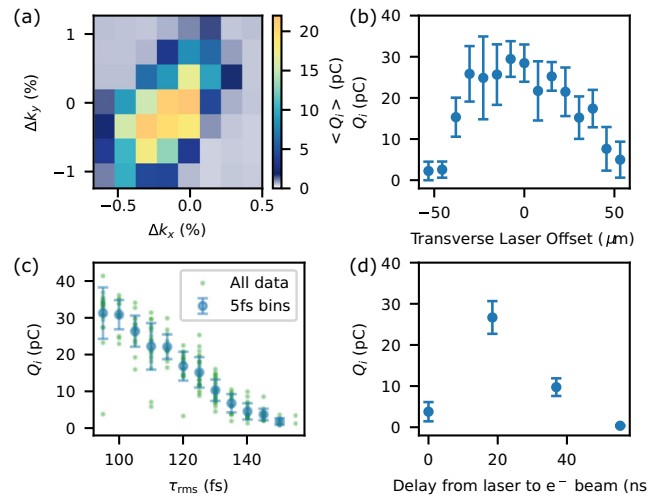

**Fig. 2 | Injected bunch optimisation. a** Mean $Q_i$ from a 2D scan of the FFQ strengths in $x$ and $y$, and $Q_i$ dependence on the transverse laser $y$ offset (**b**), driver duration (**c**) and delay between the arrival of the laser and driver $\Delta t$ (**d**). Error bars represent the standard deviation.

injected when the transverse laser was present, as also found previously[23]. The input parameters were optimised to maximise the injected charge $Q_i$ and the spectral density $dQ_i/dE$, which were strongly correlated throughout parameter scans. The four main controls over $Q_i$ and their effects are detailed in Fig. 2, where each variable was changed independently from an approximate global optimum.

In Fig. 2a, the final-focusing-quadrupole (FFQ) field strengths $k$ in the $x$ and $y$ directions were varied. Due to the large $\beta$-functions in these quadrupoles changes in their strength mostly varied the driver waist position, while changing the $\beta$-functions at focus only at the 5% level. Strength changes $\Delta k$ of 0.2–0.5% significantly altered $Q_i$, demonstrating that precise control over the bunch focusing was necessary to optimise the interaction. Simple calculations suggest that a 1% change in $k_x$, $k_y$ corresponded to focal position shifts of approximately 1 cm and 2 cm, respectively (c.f. the density profile in Fig. 1c).

Figure 2b shows the dependence of $Q_i$ on the $y$ offset of the transverse laser. $Q_i$ was strongly reduced for offsets larger than -25 μm, around half a spot width, falling to almost zero for 50 μm offsets, close to the plasma skin depth $1/k_p = 42$ μm around the density spike. Hence, this injection technique is relatively insensitive to the pointing jitter of most modern laser systems. The pointing jitter of the injection laser spot was typically 5 μm rms in each axis. While the centroid jitter of the longitudinal laser was 30–40 μm, its large spot size minimised the impact of these fluctuations.

Plotted in Fig. 2c is $Q_i$ as a function of the rms driver duration $\tau_{\rm rms}$, which was varied by altering the linac compression settings. We plot $\tau_{\rm rms}$ rather than the peak current since it was monitored non-invasively on every shot by a calibrated bunch compression monitor. $Q_i$ drops to very low levels by $\tau_{\rm rms} = 140$ fs. While bunches of such long length were not characterised with the TDS, we can simply estimate that this corresponds to $I_{\rm pk} = 1.3$ kA assuming driver charge $Q_d \propto I_{\rm pk}\tau_{\rm rms}$. Electrons can only be trapped in the wakefield if their forwards velocity, which they obtain from their interaction with the high-current driver, exceeds the velocity of the back of the wakefield in the downramp. Given the extremely steep density ramps used, it is likely that 1.3 kA is close to the minimum driver current required in practice for DDRI. Furthermore, by fitting a power-law of the form $y = A(x−b)^n$ to the data with $y = Q_i$ and $x = 1/\tau_{\rm rms}$ we retrieve $n = 0.81 \pm 0.05$. The suggested close-to-linear scaling of injected charge with drive current is promising for the extension of this scheme to higher drive currents, as the

increased driver current will be compensated almost one-to-one by increased injected bunch charge.

A final, powerful control over the injected bunch properties was the delay $\Delta t$ between the arrival of the ionising laser beams and the driver bunch, since the timing determines the height and shape of the density spike, as discussed in the next section. This was adjusted by changing the laser trigger timing in steps of 18.46 ns. $Q_i(\Delta t)$, shown in Fig. 2d, increased from a few pC at $\Delta t = 10$–20 ps to a peak of 27 pC at $\Delta t = 18.46$ ns before dropping to approximately zero over the following 37 ns. This strongly suggests that the temporal evolution of the plasma density spike affected injection, and that this process can be optimised to produce significantly higher-charge bunches.

## Optimal ramp height and length for DDRI

To understand how the laser-to-electron-bunch delay affected injection, we first consider that a plasma generated by a Gaussian laser beam is hottest in the centre, and the resulting pressure gradient drives hot plasma outwards. Initially, this decreases the height of a plasma spike. As the flow of hot plasma is supersonic, eventually a Sedov-Taylor-like shock wave with a steep, short downramp will form[36,37], which can also trigger DDRI[38,39]. Since the shock wave is unsupported its amplitude will decrease over time and eventually injection will be impossible. Imprecise knowledge of the initial conditions in this experiment precluded the use of magneto-hydrodynamics simulations, but insight in to how this general scenario affects injection was gained from a set of lab-frame quasi-3D FBPIC simulations. The purpose of this simulation set was to determine the effects of the plasma spike's height and ramp length on the injected bunch charge. The simulations were intentionally simplified to isolate the injection physics from the experimental complexities and to provide a basic physical understanding of the behaviour recorded in Fig. 2d.

The resolution of the simulations was $dz = 0.25$ μm, $dr = 0.4$ μm with 2 azimuthal modes. There were 2, 2 and 8 particles per cell in the $r$, $z$ and $\theta$ directions, except in a small longitudinal region around the injection spike, where 4, 4 and 8 particles per cell were used. The box radius was 106.3 μm and its length was 354 μm. A bi-Gaussian driver bunch was used with 304 pC charge, $I_{\rm pk} = 1.9$ kA, $\epsilon_n = 20$ mm·mrad and $\sigma_r = 16$ μm. It consisted of 300,000 macroparticles. The plasma was a 9 mm flat-top of density $n_0 = 2 \times 10^{16}$ cm$^{-3}$, with 0.5 mm linear ramps either side. At the centre, a density spike was added with a flat-top length of 40 μm, peak density $h_r n_0$, and with linear ramps of length $L_r$ either side (sketched in Fig. 3). The density varied only longitudinally, and not with radius or angle. A 2D scan of $L_r$ and $h_r$ was performed. The experimental $\Delta t = 0$ plasma spike [Fig. 1c] is well approximated by $h_r = 60$, $L_r = 30$ μm (case I). Similar results were obtained for smaller scans with flat-top lengths of 25, 10 and 2.5 μm ($1/k_p = 4.8$ μm for $h_r = 60$).

Figure 3a shows that the most charge was injected for $h_r = 8$–12, confounding the naive expectation that steeper ramps inject more charge via a slower wakefield phase velocity in the downramp. The explanation lies in Fig. 3c, which shows, at a time shortly before the arrival of the drive bunch, the longitudinal locations of the electrons that get injected, along with the density downramps in case I, and at optimal parameters $L_r = 45$ μm, $h_r = 12$ (case II). For lower $h_r$ (case II) electrons are trapped at a moderate rate throughout the downramp. For large $h_r$ (case I), a strong wakefield cannot be driven for most of the downramp because $n_e > n_b$, the bunch density, so trapping only happens close to the end of the ramp, although the higher density gradient results in rapid charge injection. In both cases, longer ramps increase the distance over which charge is trapped, increasing the injected charge. This is an unusual parameter regime, with previous works using $h_r = 1$–2[22,26] or with $L_r > 1/k_p$, for which injected charge decreases with $L_r$[21]. However, it is clearly advantageous for charge injection to have plasma-spike densities up to the drive-bunch density.

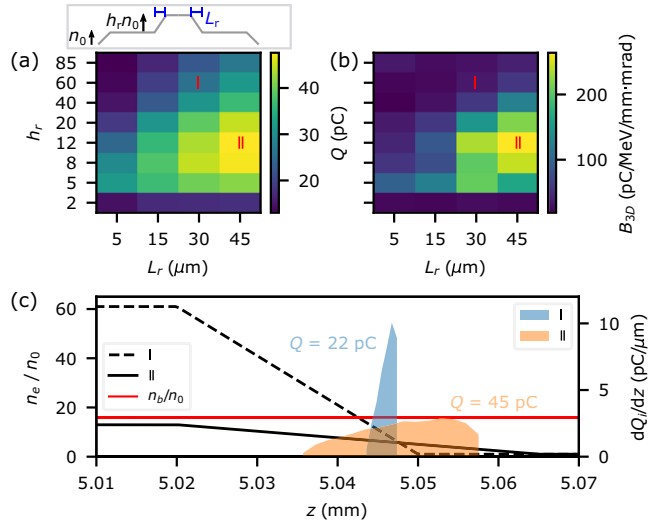

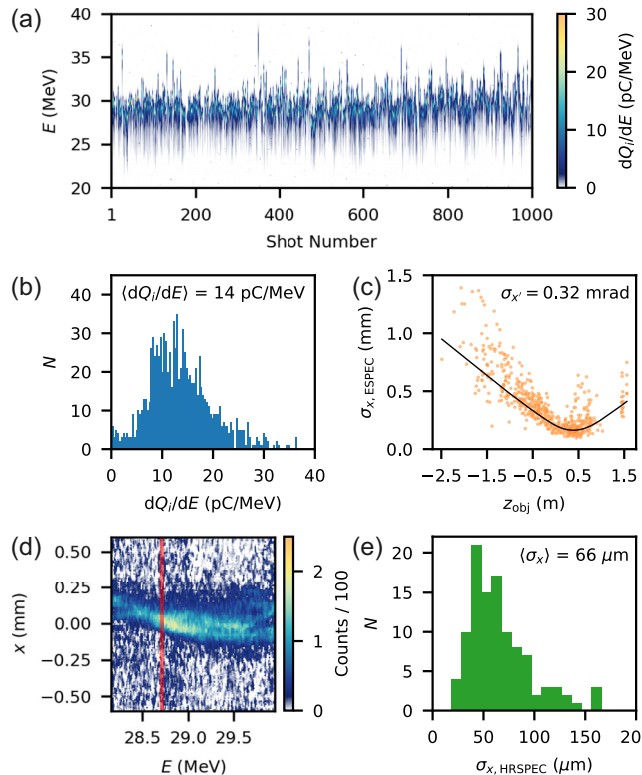

**Fig. 3 | Influence of the downramp properties on the injected bunch.**
**a**, **b** Injected bunch charge $Q$ and 3D brightness $B_{3D}$ vs. ramp length $L_r$ and ramp height $h_r$ from simulations. The inset is an illustration of the plasma density profile (not to scale). **c** Density downramp profiles (black) for cases I (experiment) and II (optimal), driver bunch density (red) and shaded areas representing histograms of the initial locations of the injected electrons for both cases.

Density downramps with peak densities $n_0$-$n_b$ maximise both the injected charge and $B_{3D}$, as shown in Fig. 3b, since the emittance varied little across this scan ($\epsilon_{n,x} = 0.4$−$0.5$ mm·mrad for $Q_i > 20$ pC). This is a surprising result, since in radiofrequency linac injectors the emittance of the bunch increases with charge, or at least current, due to space charge forces after its creation.

The hydrodynamically expanding density spike in the experiment samples a space like that in Fig. 3a. As the initially high-density peak decreases in amplitude [moving downwards from point I in Fig. 3a], the injected charge increases. At later times, as a shock front forms, the ramp length decreases (moving left or left and up) and the injected charge drops. The high charge per MeV injection resulting from these optimised density transitions, with high peak downramp densities equal to the bunch density, was a key factor in producing bright injected bunches in this experiment.

## Stability, emittance and brightness of the injected bunch

After the experimental optimisation, at the $\Delta t = 18.46$ ns data-point from Fig. 2d, a 1000-shot dataset was acquired in nominally identical conditions at 2 Hz. The injected bunch spectra, shown in Fig. 4a, were highly reproducible with a low mean $\sigma_E = 1.3 \pm 0.1\%$ FWHM, where the uncertainty is the error on the mean. The peak energies (where $dQ_i/dE$ is maximal) were tightly clustered at $(30 \pm 2)$ MeV and the mean $Q_i = (19 \pm 7)$ pC (errors are standard deviations). The mean peak spectral density was $\langle dQ_i/dE \rangle = (14 \pm 6)$ pC MeV$^{-1}$, and a histogram of all shots is shown in Fig. 4b. Its maximum value was 36 pC MeV$^{-1}$, close to that of the driver. The combination of this injection mechanism and a stable driver created a notably stable and low-energy-spread plasma-based injector.

To measure the emittance of a bunch one must obtain its width over a range of object planes, as was done for the driver. This was performed for the injected bunch on the ESPEC, shown in Fig. 4c. While providing a good divergence measurement $\sigma_{x'} = (0.32 \pm 0.01)$ mrad, it was found that low resolution combined with chromatic imaging led to a significant overestimation of the minimum bunch width $\sigma_{x,\,min}$ and therefore $\epsilon_{n,x}$. Instead $\sigma_{x,\,min}$ was measured on the HRSPEC, and $\epsilon_n$ was calculated in the absence of the correlation term i.e. $\langle \epsilon_{n,x} \rangle = \gamma\beta\langle\sigma_{x,\,min}\rangle\langle\sigma_{x'}\rangle$. Bunch jitters over the long distance to the HRSPEC combined with the small field-of-view of the diagnostic made it

**Fig. 4 | Analysis of the optimised injected bunch from the experiment.**
**a** Waterfall plot of 1000 consecutive energy spectra. **b** Histogram of peak spectral density $dQ_i/dE$ from the same bunches. **c** Injected bunch widths (orange) vs. object plane measured by the electron spectrometer, with a fit shown in black. **d** Example high-resolution electron spectrometer image. The red line indicates the bin used to determine the bunch width. **e** Histogram of bunch widths ($\sigma_{x,\,min}$) measured on the high-resolution electron spectrometer.

impractical to perform an object-plane scan. Nevertheless, 106 measurements of $\sigma_{x,\,min}$ were made. An example image is shown in Fig. 4d. Both the bunch tilt and defocussing from chromatic effects (e.g. beyond 29.5 MeV) meant that an energy slice rather than the projection gave a fair measurement of $\sigma_{x,\,min}$. Details of the $\sigma_{x,\,min}$ calculation from the HRSPEC images are given in the 'Methods' section. A histogram of measured $\sigma_{x,\,min}$ is shown in Fig. 4d. The mean value with its standard error was $\langle\sigma_{x,\,min}\rangle = (66 \pm 3)$ μm, which was significantly above the resolution limit of 2.5 μm. Combining this with the divergence $\langle\epsilon_{n,x}\rangle = (1.2 \pm 0.1)$ mm·mrad was found. Using the mode $\sigma_{x,\,min} = 43$ μm, $\epsilon_{n,x} = 0.8$ mm·mrad. The 3D brightness $B_{3D} = \langle dQ_i/dE \rangle / \langle\epsilon_{n,x}\rangle = 11.3$ pC/MeV/mm·mrad. Note that this value is constructed only from averaged values, with the exception of the injected bunch divergence, which was determined from a fit to a scan including ~600 data points. In this experiment the driver $B_{3D,d} = 2.4$ pC/MeV/mm·mrad.

## 6D brightness estimate from particle-in-cell simulations

In the experiment, it was not possible to measure $\epsilon_{n,y}$ or the current profile of the injected bunch, so to estimate 6D brightness ($B_{6D}$) PIC simulations were required. Figure 5 shows the longitudinal phase space, current profile and slice emittances of the injected bunch from the end of a lab-frame, fully 3D WarpX simulation[40], which used cell sizes $dx = dy = 0.39$ μm, $dz = 0.14$ μm, a box size of $200 \times 200 \times 400$ μm and 1 and 4 particles per cell in the background plasma and plasma spike regions, respectively. The longitudinal plasma density was that from Fig. 1c, which was our best estimate of the plasma density from the experiment, as described in the Methods section. In $y$ the spike density profile was that from Fig. 1c inset (i.e. at $\Delta t \approx 0$), and in $x$, the transverse laser propagation direction, it extended infinitely. The

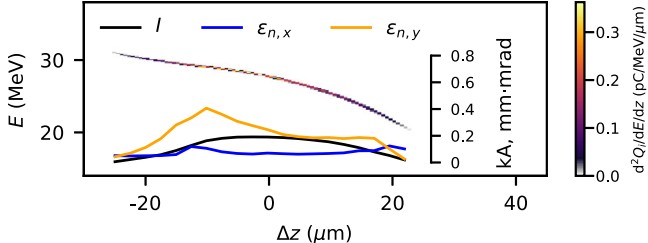

**Fig. 5 | Simulated phase space of the injected bunch.** The longitudinal $E-z$ phase space, normalised slice emittances in $x$ and $y$ and the current profile of the simulated injected bunch.

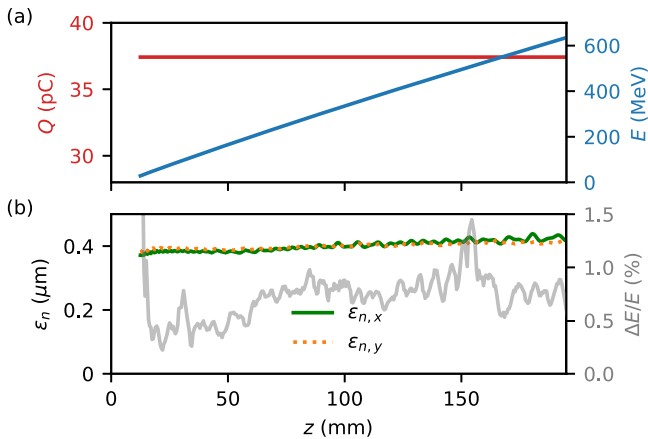

**Fig. 6 | Achieving higher energy gains.** Results from the 195 mm constant plasma density simulation with a density spike for injection at $z = 5$ mm, showing the evolution of (**a**) the injected bunch charge and energy, and **b** the normalised emittance in both axes and the relative energy spread.

driver current profile from Fig. 1b) was used along with a transverse emittance and bunch width in both directions of 20 mm·mrad and 16.5 μm ($\sigma$ for a Gaussian bunch with the measured emittance and beta function). Figure 5a shows that the injected bunch, consisting of $1.3 \times 10^5$ macroparticles, was accelerated to a peak $E = 28.6$ MeV, consistent with the experiment. The injected charge was 19.3 pC and the peak $dQ_i/dE = 5.7$ pC/MeV. The slice emittances are asymmetric, as are the projected $\epsilon_{n,x} = 0.14$ mm·mrad and $\epsilon_{n,y} = 0.55$ mm·mrad, suggesting that the curvature of the plasma spike profile in $y$ caused a larger emittance. This could be partially mitigated by using a larger transverse laser spot or an astigmatic focus[41]. The simulated $B_{3D} = 42$ pC/MeV/mm·mrad, calculated using the projected emittances, was 3.7 times the experimental value, which used a conservative measurement of $\epsilon_{n,x}$. The injected bunch peak current was 0.19 kA, resulting in $B_{6D} = 1.3$ kA MeV$^{-1}$ mm$^{-2}$ mrad$^{-2}$.

## Discussion

We now consider how to improve on the experimental results, where the lower $\epsilon_n$ of the injected bunch came at the price of low electron energy, which would be unfortunate for short-wavelength light-source focussed applications. This was, however, purely a result of suboptimal plasma creation due to ionisation defocusing of the longitudinal laser pulse. As an example improvement we consider a 195 mm constant density plasma at $n_e = 2 \times 10^{16}$ cm$^{-3}$, which can be created via discharge ionisation. Then a short pulse laser requiring only $\mathcal{O}(10)$ mJ energy per pulse can be used to create an injection spike with a density of 12 times the background plasma density by using a gas mixture of 9 parts helium to 2 parts argon, reasonably assuming that the discharge ionises only the first level of argon. The use of an electron driver, discharge ionisation and relatively low energy laser pulses would allow the repetition rate of a similar experiment to be pushed in to the MHz regime. MHz pulsers and burst-mode lasers are likely to be able to accommodate the bunch patterns of superconducting RF linacs[42]. The case II simulation from Fig. 3 was extended in FBPIC to a 195 mm long plasma using the same drive bunch. Figure 6a shows the injected bunch energy and charge, while Fig. 6b plots the evolution of the projected $\epsilon_n$ and relative energy spread. The injected charge remains constant throughout the plasma at 37.4 pC, while only a small growth in $\epsilon_n$ in both axes is observed to just over 0.4 mm·mrad. The peak electron energy increases linearly with propagation distance to 635 MeV at $z = 195$ mm at a rate of 3.3 GeV m$^{-1}$. The FWHM energy spread of the bunch at $z = 195$ mm remains low at 0.6%, but is increased in absolute terms from a minimum value of 0.1 MeV shortly after injection to 3.7 MeV at the end of the plasma due to improper beam loading[5,43]. This may be improved by shaping the downramp profile via alterations made to the laser spot shape using an adaptive optic, or by using an escort bunch[8].

This scheme is scalable to higher injected-bunch brightness, which is predicted to increase with plasma density as $B_{6D} \propto n_e$ since, for the same driver current, the transverse momentum of injected electrons is constant while the spatial scales shrink with the plasma

wavelength $\lambda_p \propto 1/\sqrt{n_e}$ in each dimension[18]. In the near future driver bunches as short as 1.7 fs should be available[44], compared to a 96 fs-long bunch used in this demonstration. Keeping the bunch length $\tau \propto \lambda_p \propto 1/\sqrt{n_e}$, the density, and thus 6D brightness, could be increased over 3000-fold thanks to a 56 times reduction in emittance in both planes. A properly scaled version of this experiment could, therefore, be a source of exceptionally bright femtosecond electron bunches with normalised emittances of only a few tens of nanometres.

In summary, we have presented experimental results from a beam-driven plasma accelerator in which a high-quality electron bunch was injected via density-downramp injection. The 1.3% energy spread, 14 pC MeV$^{-1}$ bunches were accelerated at approximately 1 GV m$^{-1}$ with high reproducibility. The mean normalised emittance of the injected bunch was measured to be $(1.2 \pm 0.1)$ mm·mrad. The combination of high charge per MeV and low emittance was a result of using a short spike in the plasma density profile, with a peak density an order of magnitude greater than the surrounding plasma density. Particle-in-cell simulations showed that peak spike densities up to the drive beam density increased both the charge and brightness of the injected bunches.

## Methods
### Plasma profile
The plasma was created in a 50 mm long, 1.5 mm diameter open-ended capillary, which was continuously filled from a 46 mbar buffer containing 97% Ar and 3% H$_2$ by volume. The H$_2$ dopant was used as part of an offline line-broadening-based density diagnostic. The gas density profile (Fig. 1c) was estimated using a benchmarked ANSYS Fluent model. The plasma was laser-ionised with $(32.6 \pm 0.2)$ fs full-width-half-maximum (FWHM) duration pulses from a Ti:Sapph system. The longitudinal laser pulse, which ionised the bulk of the plasma, had a FWHM spot size of $(430 \pm 10)$ μm $\times$ $(350 \pm 10)$ μm and a peak intensity of $(3.0 \pm 0.1) \times 10^{14}$ W cm$^{-2}$. The transverse laser was focussed to a spot size of $(48 \pm 1)$ μm $\times$ $(43 \pm 1)$ μm FWHM and intensity of $(5.2 \pm 0.1) \times 10^{15}$ W cm$^{-2}$ (all uncertainties here are the standard deviation). The plasma electron density ($n_e$) profile in Fig. 1c was produced by an FBPIC simulation[45,46] employing an ADK ionisation model[47], using the simulated gas profile and measured laser parameters.

### Drive bunch emittance measurement
The emittance of the drive bunch was found by measuring its rms width $\sigma_x$ throughout a scan of the downstream quadrupole triplet's object plane $z_{obj}$. The emittance was retrieved from the fit to

$$\sigma_x(z_{obj}) = \sqrt{\frac{\epsilon_{n,x}}{\gamma\beta^*}[\beta^{*2} + (z_{obj} - z_0)^2/\beta^*]},$$ where $\beta^*$ and $\gamma$ take their usual

definitions from special relativity, $z_0$ is the virtual source point and $\beta^*$ is the beta-function at $z_0$. An example of this method is shown in Fig. 1d.

**Electron and laser beam spatio-temporal overlap**

The longitudinal laser was overlapped spatially with the electron bunch using two optical-transmission-radiation screens up- and downstream of the capillary, imaged onto CMOS cameras. The upstream camera also imaged the focal spot of the transverse (injection) laser, which was translated to the same height as the electron beam. The laser and electron beam temporal overlap was found to within 0.5 ps using the plasma afterglow light technique[48], whereby the interaction of an electron bunch with a pre-formed plasma produces strong light emission. Both laser beams were derived from the same pulse, and their synchronisation with the electron bunch was maintained with < 0.1 ps jitter using the system described in ref. 49. Changes in the amplitude of the plasma glow signal with position of both laser beams provided an additional check on the spatial overlap.

**Calculation of the minimum bunch width from the HRSPEC images of the injected bunch**

The chief difficulties in determining $\sigma_{x,\,min}$ from these images were the presence of hot pixels (i.e. single or small clusters of pixels with high counts, which can be formed via the detection of a high-energy bremsstrahlung photon, for example) and the low signal-to-noise ratio. To stop the fitting algorithm from erroneously identifying a hot pixel as a narrow beam slice, the hot pixels were removed with a $3 \times 3$ median filter. It could then be the case that the remaining low-level noise, or part of it, could be identified by the algorithm as part of the bunch. To remove this possibility, the image was first binned into 11-pixel-wide slices along the energy axis to increase the signal-to-noise ratio. Then, a region of the binned image far from the signal was used to find the maximum per-pixel noise counts $C_n$, and bins with maximum values $< N_{min} C_n$ were rejected from the analysis. This ensured a high level of confidence that the remaining image slices contained a high signal level attributable to the injected electron bunch. In practice $N_{min} = 2.5$ provided a good balance between a low level of image rejection and eliminating erroneous fitting to noise. Gaussians were fitted to the remaining slices, and the smallest width was selected as the best estimate of the true bunch width. A manual check was performed to ensure that the fit was to a slice containing part of the injected electron bunch, giving further confidence in the image processing and fitting algorithm. While the median filter kernel size, bin width and $N_{min}$ were free parameters, varying each of them by a factor of two changed the mean measured emittance by <10%. The median filter, binning and the elimination of low-signal slices all act to increase the retrieved electron bunch width, meaning that our stated brightness represents a conservative measurement.

## Data availability

The data and plot scripts for all figures are freely available in the Zenodo repository at https://doi.org/10.5281/zenodo.17964753[50].

## Code availability

The input scripts used in the simulations for this study are available at https://doi.org/10.5281/zenodo.17964753[50]. The codes FBPIC and WarpX are open source and freely available.

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

## Acknowledgements

The authors thank M. Dinter, S. Karstensen, S. Kottler, K. Ludwig, F. Marutzky, A. Rahali, V. Rybnikov and A. Schleiermacher for their engineering and technical support. They additionally thank M. Meisel and T. Staufer for sharing their laser expertise and M. Kirchen for helpful discussions on PIC simulations. This work was supported by the Maxwell computational resources at DESY. The authors gratefully acknowledge the Gauss Centre for Supercomputing e.V. (www.gauss-centre.eu) for funding this project by providing computing time through the John von Neumann Institute for Computing (NIC) on the GCS Supercomputer JUWELS at Jülich Supercomputing Centre (JSC). This research used the open-source particle-in-cell code WarpX https://github.com/ECP-WarpX/WarpX. This work was supported by Helmholtz ARD, Helmholtz ATHENA, the Helmholtz IuVF ZT-0009 programme and the Maxwell computational resources at DESY. L.B. was supported by an Engineering & Physical Sciences Research Council (EPSRC) Studentship.

## Author contributions

J.C.W., L.B., J.B., J.B.S., S.B., J.M.G., P.G.C., C.A.L., G.L., F.P., S.S. and R.D. performed the experiment with significant help from from K.P and S.W. G.B., J.M.G., S.M.M., T.P. and M.T. contributed to the understanding of the plasma density profile. J.C.W. and L.B. upgraded and maintained the laser beamline and related diagnostics, with help from S.B. and K.P. J.C.W. analysed the data and produced the figures, with significant input from L.B., and wrote the manuscript. The particle-in-cell simulations in Figs. 3 and 6 were performed and analysed by J.C.W., while the simulation in Fig. 5 was performed and analysed by L.B with input from M.T. R.D. and J.O. supervised the personnel and the project. The experiment was originally conceived by R.D. and J.O. All authors discussed the results in the paper.

## Funding

## Competing interests

The authors declare no competing interests.
