## [Transparent Peer Review file · Nature Communications]

Bright Electron Bunches from a Plasma-Wakefield Accelerator with a Steep Density Down-Ramp

Corresponding Author: Dr Jonathan Wood

Version 0:

REVIEWER COMMENTS

Reviewer #1 (Remarks to the Author):

Referee report on Brightness-Boosted Electron Bunches from a Plasma Accelerator, by J.C. Wood et al.

The manuscript deals with experimental results, about the generation of high brightness electron beams via PWFA driven by a LINAC, with “witness” (accelerated) electron beams obtained by density downramp injections. The results reported in the manuscript are interesting for the plasma acceleration community, the discussed physical processes are generally sound and the manuscript is well written. As there are, however, some specific points to be clarified or where the manuscript can be improved, the Referee suggests Authors to revise their manuscript accordingly to the items below.

Question1: Pg4, left column, after “...the plasma was a 9 mm flat-top ..”.

Why Authors used a flat-top density profile for their simulations although the plasma profile (by their simulations) is decreasing [see Fig 1c] ? The same question can be raised for the WarpX simulation shown after. Can Authors comment on this?

Question2: As the energy of the accelerated beam is quite low, is the value of measured emittance mostly getting contributions from the injection/acceleration or from the downramp?

Authors did q3D and 3D simulations but they didn't report on the evolution of the phase longitudinal and transverse space cuts along the beams propagation. This would definitely help in understanding how the energy spread compression is made (the superposition with the longitudinal on-axis gradient will help) and how the x,y emittances vary with the longitudinal position. It's also important to distinguish between the B_{3D} at the end of the acceleration and that after the downramp, in order to have a view on the room they have for minimizing the emittance growth in the downramp. The Referee suggests to add all or at least some of the relevant information reported above.

Minor changes:

The title looks somewhat misleading. The word “boosted” recalls a (large) increase of a quantity which is already existing. For example, we can safely name ‘brightness-boosted bunches’ any bunch having a certain brightness before the acceleration stage and having higher brightness at its end. In the case of the present manuscript title, Authors claim they created ‘brightness-boosted’ bunches as the final brightness of those bunches exceeds that of the wakefield driver. My suggestion is to change the title by referring to high brightness electron beams. The last-last sentence of the Abstract should be changed accordingly.

1. Pg3, left column. “.In Fig. 2(a) the final-focussing-quadrupole (FFQ) field strengths k in the x and y directions, which almost exclusively determined the driver waist position in z while preserving its spot size, were varied”

Authors should add information about the longitudinal variation of the driver beam focus once the field strength is changed. This, along with the information of the beam beta function, will give direct information on how sensitive is the PWFA process on the drive final focus positions, with the optimized parameters of this experiment. It would be helpful to give these spatial information together with the density map profile, which also plays a relevant role.

2. Pg3, right column after “Given the extremely steep density ramps used in this work we suggest this as being close to the minimum peak current required to trigger DDRI in PWFAs”. The referee definitively agrees with the analysis, but the sentence should be made more clear. It may help the introduction of the concept of minimum wakefield amplitude for sharp

density downramps.

3. Pg3, right column after "This was adjusted by changing the laser trigger timing in steps of 18.46 ns. $Q_i(\Delta t)$, shown in Fig. 2(d), increased from a few pC at $\Delta t = 10\text{--}20$ ps to a peak of 27 pC at $\Delta t = 18.46$ ns before dropping to approximately zero over the following 37 ns".

As this is one of the key points of the density downramp scheme, the reader expects a sentence here about the time evolution of the electron density spike generated by the ionization laser pulse. At least Authors can add "(...as the timing is linked to the electron density spike profile, as shortly discussed in the next paragraph)"

4. Pg4, left column, after "The resolution was $dz = 0.25$ μm , $dr = 0.4$ μm with 2 azimuthal modes". Authors should also include information about the box (cylinder) radius, length, the number of ppc ($N_r \times N_t \times N_z$) of the plasma and how they simulated the driver (as a distribution, with given macroparticles) and, most importantly, if they used a Lorentz boosted frame technique (in the case, as that technique also squeezes the plasma, they should also add information about the total number of macroparticles describing the accelerated beam, information which is relevant to understand the quality of the phase-space analysis)

5. Pg5, right column. Related to the WarpX simulation, Authors should state the number of macroparticles which result in the accelerated beam (as for FB-PIC: did the use a Lorentz boosted technique?). Did Authors use impact ionization for the plasma?

Reviewer #2 (Remarks to the Author):

The manuscript by J. C. Wood et al., titled "Brightness-boosted electron bunches from a plasma accelerator" describes the first experimental demonstration that achieved both low emittance and energy spread. As described in the manuscript, previous experimental demonstrations only achieved a large emittance (10 micron) and a few micron emittance with a large energy spread respectively. However, this demonstration achieved a 1 micron emittance with ~ 30 pC charge. Also, the spectral density has improved by almost a factor of 10 compared to the previous researches. While these numbers are still much worse than what we can generate using conventional linacs, it is remarkable progress in this new scheme, which has big potential to overcome the limitations that conventional methods have. Furthermore, authors provided experimental study on system parameters that turned out to be critical for the total charge generated and explained the principle behind it using simulations. Such observations would significantly help subsequent researches on plasma-based electron sources. I believe the demonstrated results and analysis in this manuscript are indeed an important step towards the realization of plasma-based high-brightness electron sources, which are critical for future accelerators. Its impact would naturally be broad. Therefore, I believe this manuscript satisfies Nature Communications' publication criteria. However, there are two major issues and several minor items that must be addressed by authors.

Major issue:

[1] The authors claim that the scheme they presented made a significant boost in brightness. However, this claim is not accurate. The drive beam used in the experiment had a charge of ~ 300 pC, and its emittance was 20 micron. This emittance is almost 40 times larger than what I would expect from a 300 pC charge. If a nicely generated drive beam of the same charge has been prepared, the brightness of the drive beam could be improved by a factor of 1600. It is unfair to claim enhancement in brightness using such low-quality drive beam. Additionally, to claim that the scheme is really improving the brightness, such transverse emittance improvement on the drive beam must improve the main beam's up to some level. However, based on the description in this manuscript, I don't expect such improvements. Furthermore, the achievable brightness usually improves as the charge decreases. Thus, directly comparing the driver and injected bunches, having different charge level, is inappropriate. If we call such scheme as brightness boost, almost any schemes using two beams with difference charges can be considered as brightness boost. The wording "brightness boost" is incorrect and inappropriate. It must be eliminated for publication. The results of this manuscript are solid. I suggest that authors focus on their great demonstration of high-brightness plasma-based electron source.

[2] It was not clear to me what changes were made from other researches and what led to the improvements presented by authors. The authors need to provide more explanations regarding the changes they made compared to previous researches and how those changes contributed to achieving the enhancements.

Minor items:

1. The introduction doesn't seem nicely organized. For example, the first two paragraphs almost repeat the same argument. The third one focuses on emittance and what can be possibly achieved, which is ok. However, the fourth paragraph goes back to why you need DDRI and beam-driven. Also, the fourth paragraph explains too much low level details without any other explanations. It was hard to read and hard to catch what authors wanted to say. I think such difficulty was coming from the paragraph and sentence structures. It would be great if the authors can improve this section.

2. "as they approach the back of the wake they experience a magnetic field from the radial current of sheath electron which act as a defocusing force". Is it defocusing force not focusing force?

3. It would be good to have more explanation of DDRI in the introduction. Currently it is forcing readers to find references to understand what that is, and it seems inappropriate for Nature Communications' readers. It doesn't have to be long but would be nice if you have what it is and why you do it (or what advantages are expected).

4. "the drive waist position in z while preserving its spot size". If you change the focusing point in z direction, it changes the spot size. Do you mean by the change is ignorable? Or you actually made other compensation upstream? It would be better

to describe it more precisely.

Reviewer #3 (Remarks to the Author):

The article "Brightness- Boosted Electron Bunches from a Plasma Accelerator" describes an electron driven Plasma wakefield acceleration experiment using internal injection of the witness bunch. In injection mechanism is a density downramp injection.

The main result is a 3D brightness measurement of the accelerated witness bunch which the authors claim is 4.8 times higher than the originally driving electron bunch.

The paper is well written, rather clear and definitely original. The results presented are new and certainly important for the field of plasma acceleration. The paper describes a complicated experiment combined with a challenging data analysis. I therefore recommend the paper to be published. I would like the authors to consider nevertheless the following comments:

Introduction:

To my opinion the authors describe somewhat inaccurate the limitation of conventional accelerators. For example a state of the art FEL is only a few 100 m long not several kilometers (for example Swiss FEL). Furthermore those conventional FEL's produce beams with an emittance of 100 nm for 100 pC charge. Therefore actually superior of what has been achieved in this paper. See for example:

E. Prat, P. Dijkstal, M. Aiba, S. Bettoni, P. Craievich, E. Ferrari, R. Ischebeck, F. Löhl, A. Malyzhenkov, G. L. Orlandi, S. Reiche, and T. Schietinger Phys. Rev. Lett. 123, 234801, 2019. DOI:

This facts should be mentioned which does not diminish the achievements of the publication.

I find the term brightness-boost extremely misleading because there are two different beams, one loses energy and the other gains energy and is actually generated within the plasma. What is actually the brightness of the witness beam at its source if this can be simulated/determined.

Results:

In this section I am missing the detailed data of the result.

Which bunch charge, which emittance and energy spread was achieved. An image of both beams would be nice as well. It comes later but those are major results.

Optimal Ramp Height & Length for DDRI:

This chapter is somewhat confusing. It seems to be a simulation study but it is not so clear how it is related to the experiment.

Did the experiment had clear control on the ramp profile, do we know which profile gave the best result in the experiment. ? Please clarify this points.

6D brightness estimate from PIC simulations:

The authors like to suggest that the brightness boost in 6D could be more impressive than the measured one in 3D. The simulations seem to show quite asymmetric emittance values, wouldn't this suggest that the 3D measurement presented before might be optimistic because likely measured in the lucky plane?

Stability, Emittance and brightness of the injected bunch:

It would be nice to make clearer what results are averaged over the data set and which are best shots results in this chapter. Is the 4.8 times boost a best shot result ? What is the average of the data sample ?

Reviewer #4 (Remarks to the Author):

The paper presents an experimental investigation of high-brightness electron bunches generation from a plasma accelerator, achieved through injection control of improved density transition using an additional injection laser to generate a higher density plasma spike. The injected electron bunch exhibits an improved brightness, with a 3D brightness 4.8 times greater compared to the driver electron bunch. The experimental results are elaborately discussed alongside particle-in-cell simulations. However, it is noted that a brightness-boosting scheme was previously proposed in [Phys. Rev. STAB. 7. 011301 (2004)] (Ref. [16] in the paper), and the experimental setup is almost identical to their previous work (Phys. Rev. Accel. Beams. 24. 101302 (2021)). As such, this paper appears to be more of an experimental verification work rather than one that extends the boundaries of knowledge. Therefore, it may not be suitable for publication in Nature Communications.

There are several additional issues that need further discussion and revise:

1. In the scan simulations using the quasi-3D FBPIC code, which employs a cylindrical coordinate model, the simulated system needs to exhibit cylindrical symmetry. However, the spike plasma density created by the tightly focused injection laser deviates from cylindrical symmetry, which could impact the accuracy of the simulations.
2. The authors mention that, for the experimental results, although the emittance of the injected electron bunch is 17 times lower than that of the driver, the 3D brightness (B3D) only shows an increase of 4.8 times. This discrepancy is attributed to the deduced dQ_i/dE , which decreases as the energy spread increases. This challenge highlights the need to control the energy spread and phase space rotation of the injected electron bunch.
3. The authors suggest that their simulations indicate a potential >1000-fold increase in the 6D brightness (B6D) in their

experimental configuration. However, it is noted that the emittance values used in the simulations are slice-normalized emittances, whereas the experimental results provide normalized transverse emittance values. This discrepancy could be misleading, as slice emittance is typically much lower than normalized transverse emittance. Furthermore, the simulations do not include the parameters of the driver, and it would be beneficial to compare the phase space distribution between the driver and the injected electron bunch.

4. In Figure 6(b), the emittance appears different from that in Figure 5. It would be beneficial to include the brightness evolution in this figure for a more comprehensive analysis.

5. The format of the references should be revised. For example, in reference [10], "Plasma Physics and Controlled Fusion" should be formatted as "...Plasma Phys. Control. Fusion..."

Version 1:

Reviewer comments:

Reviewer #2

(Remarks to the Author)

As already commented in earlier report, I believe this manuscript satisfies Nature Communications' publication criteria.

The previous report provided two major issues and a few minor issues that must be addressed. Authors provided appropriate replies to each issue and updated the manuscript accordingly.

I recommend the publication of this manuscript.

Reviewer #3

(Remarks to the Author)

Revision of the paper "Bright Electron Bunches from a Plasma-Wakefield Accelerator with a Step Density Down-Ramp" by J.C. Wood et al.

The authors prepared a very detailed and rigorous response to the reviewers questions and concerns.

They made a number of changes in the manuscript in order to address the issues raised by the reviewers.

I am happy with the revised paper and feel my concerns have been taken into account appropriately.

I recommend to publish the paper in this version.

DESY | Jonathan Wood, Notkestraße 85, 22607 Hamburg

Jonathan Wood
Teamleader for Beam Driven Plasma
Accelerators
jonathan.wood@desy.de
Notkestraße 85
22607Hamburg
Tel. +49 40-8998-5297

6th October 2025

Response to the Reviewers

Dear Editor,

On behalf of all of the co-authors, I would like to thank you for the continued consideration of this manuscript and to thank the reviewers for taking their time to engage with and provide their valuable feedback on this work. We apologise for the delay in the production of this response, although we now believe that we have managed to engage with and answer every one of the concerns raised by each of the reviewers. In the rest of this letter a response to the reviewer's comments is provided, with their original comments in **blue text**, our response in black, and changes made to the manuscript in **green**. Unless stated otherwise, line numbers refer to locations in the updated rather than the original manuscript.

Reviewer 1

The manuscript deals with experimental results, about the generation of high brightness electron beams via PWFA driven by a LINAC, with "witness" (accelerated) electron beams obtained by density downramp injections. The results reported in the manuscript are interesting for the plasma acceleration community, the discussed physical processes are generally sound and the manuscript is well written. As there are, however, some specific points to be clarified or where the manuscript can be improved, the Referee suggests Authors to revise their manuscript accordingly to the items below.

We thank the reviewer for their complimentary remarks and their interest in this work. Individual responses to their suggestions are made below.

Question1: Pg4, left column, after "...the plasma was a 9 mm flat-top ..". Why Authors used a flat-top density profile for their simulations although the plasma profile (by their simulations) is decreasing [see Fig 1c] ? The same question can be raised for the WarpX simulation shown after. Can Authors comment on this?

Deutsches Elektronen-Synchrotron DESY
Notkestraße 85, 22607 Hamburg, Germany

Location Zeuthen
Platanenalle 6, 15738 Zeuthen, Germany
www.desy.de

Board of Directors

Prof. Dr. Dr. h. c. Beate Heinemann
(Chairperson)
Dr. Ties Behnke (interim)
Prof. Dr. Wim Leemanns

Prof. Dr. Britta Redlich
Prof. Dr. Christian Stegmann
Iris Wilhelm
(Deputy Chairperson)
Dr. Arik Wilner (CTO)

Figure 2(d) of the manuscript strongly suggests that the evolution of the downramp over time dramatically changes the amount of injected charge, and we postulate based on known physics that this is likely to be a result of the evolving height and length of the density downramp. The purpose of the FBPIC simulations was to explore how different ramp heights and lengths affected the charge, and incidentally the brightness, of the injected bunch. For this study we found it beneficial to simplify the problem by taking a simplified plasma density profile. While this did not accurately represent the experiment, it provided great insight in to why the injected charge was low for a small laser-to-electron-beam delay, and suggested an explanation for the observed behaviour in Figure 2(d). Furthermore, these simulations showed that it can be beneficial to use large ramp heights, with much higher relative densities than have been used in literature before, and we explained this behaviour. In order to clarify the purpose of these simulations in the manuscript, we have added the following text to lines 261-267:

“The purpose of this simulation set was to determine the effects of the plasma spike’s height and ramp length on the injected bunch charge. The simulations were intentionally simplified to isolate the injection physics from the experimental complexities, and to provide a basic physical understanding of the behaviour recorded in Fig. 2(d).”

The full-3D WarpX simulation was very computationally expensive, and it was not possible to perform parameter scans at sufficiently high resolution. The WarpX simulation did not use a flat-top plasma density profile, it used the plasma profile shown in figure 1(c) of the manuscript, which was our best estimate of the experimental plasma density profile. This was originally stated in the section “6D Brightness Estimate from Particle-in-Cell Simulations” with the line “ $n_e(z)$ from Fig. 1(c) was used”. This was too brief and made it too easy to miss this important point. This line was replaced with the following text on lines 390-394 of the manuscript:

“The longitudinal plasma density was that from Fig. 1(c), which was our best estimate of the experimental plasma density from the experiment, as described in the Methods section.”

Question2: As the energy of the accelerated beam is quite low, is the value of measured emittance mostly getting contributions from the injection/acceleration or from the downramp?

It was unfortunately not possible to determine this in the experiment, since we can only measure what makes it to our detectors, and had little control over the overall longitudinal plasma density profile, particularly the long downramp in to the vacuum section. In the figure below, we plot the evolution of the emittance of the injected bunch in the WarpX simulation in both x and y , as well as the simulated plasma density profile.

Evolution of the normalised emittance of the injected beam in the WarpX simulation detailed in the manuscript. The shaded blue region is the plasma density profile (note the logarithmic scale).

The emittance of the bunch does not change during the transition from plasma to vacuum. At the injection point ($z = 20$ mm) the emittance of each bunch is surprisingly similar, and at the 100 nm level. While the emittance rapidly saturates in the x direction, in the y direction it continues to grow. Typically this is due to betatron decoherence, with badly matched bunches experiencing greater emittance growth during acceleration in the plasma. It is, thus, strongly indicated by the simulations that the emittance after

the plasma, which is what was quoted in the manuscript, has its value as a result of the injection and acceleration process, rather than being an effect of the plasma-to-vacuum transition.

Authors did q3D and 3D simulations but they didn't report on the evolution of the phase longitudinal and transverse space cuts along the beams propagation. This would definitely help in understanding how the energy spread compression is made (the superposition with the longitudinal on-axis gradient will help) and how the x,y emittances vary with the longitudinal position. It's also important to distinguish between the B_{3D} at the end of the acceleration and that after the downramp, in order to have a view on the room they have for minimizing the emittance growth in the downramp. The Referee suggests to add all or at least some of the relevant information reported above.

First regarding the transverse phase space. In the q3D (FBPIC) simulations, the emittance in both planes was approximately constant in both planes after injection. This can be seen in Fig. 6 of the manuscript, which is a repeat of the case-II simulation from Fig. 3(a)&(b) but with a longer plasma after the injection point. In the full 3D (WarpX) simulation, we hope that the answer to the previous question assuages the reviewer's concerns.

Regarding the longitudinal phase space, below we plot the evolution of the absolute and relative energy spreads in the full 3D simulation. While the simulation shows that a bunch with low emittance in both planes can be injected via our method, the simulation significantly overestimates the energy spread with respect to the experiment. The agreement in the total energy gain between the simulation and experiment suggests that our understanding of the plasma profile post-injection is good, so we attribute this disagreement to a lack of knowledge of the precise shape of the density spike. This could not be measured in the experiment- it would have required diagnosis of a very underdense plasma with micrometre resolution, which is not possible in our accelerator environment. The WarpX simulation was a best-effort attempt to recreate the $\Delta t = 0$ data point from the experiment. We believe that it adds value to the paper by showing that in principle bunches can be injected with significantly sub-micrometre normalised emittances in both planes. However we would prefer not to add the longitudinal analysis of the simulation to the manuscript since it agrees poorly with the data. We lack an understanding of how the energy spread compression happens in the experiment, and altering the 3D shape of the density spike to reduce the energy spread towards the experiment value would be both speculative and an ill-posed problem, since several features could be changed with only a single result to compare to. For the longer q3D simulation in figure 6 we have added the (unoptimised) energy spread to the plot. This shows that, in principle, bunches injected via our technique can maintain a small absolute energy spread and approximately constant relative energy spread while experiencing a large energy gain. As such it is not always the case that a large compression of the energy spread is required.

Evolution of absolute and relative energy spreads in the WarpX simulation.

Minor changes:

The title looks somewhat misleading. The word "boosted" recalls a (large) increase of a quantity which is already existing. For example, we can safely name 'brightness-boosted bunches' any bunch having a certain brightness before the acceleration stage and having higher brightness at its end. In the case of the

present manuscript title, Authors claim they created 'brightness-boosted' bunches as the final brightness of those bunches exceeds that of the wakefield driver. My suggestion is to change the title by referring to high brightness electron beams. The last-last sentence of the Abstract should be changed accordingly.

We accept this comment from the reviewer. The concept of producing a high-brightness electron beam in a plasma accelerator driven by a low-brightness electron beam has been coined a 'brightness booster' within the plasma wakefield accelerator community, although as pointed out this name can be misleading. We have changed the title to: 'Bright Electron Bunches from a Plasma-Wakefield Accelerator with a Steep Density Down-Ramp'. We also removed the phrase brightness booster from the abstract, and the final sentences of it now read 'Plasma accelerators therefore offer a possibility to create high-brightness bunches in wakefields driven even by low-quality drive bunches. Here we demonstrate the injection and gigavolt-per-metre acceleration of electron bunches with mm-mrad normalised emittance, $\mathcal{O}(10\text{ pC/MeV})$ spectral density and per-cent-level energy spread, all with excellent reproducibility.'

1. Pg3, left column. ".In Fig. 2(a) the final-focussing-quadrupole (FFQ) field strengths k in the x and y directions, which almost exclusively determined the driver waist position in z while preserving its spot size, were varied" Authors should add information about the longitudinal variation of the driver beam focus once the field strength is changed. This, along with the information of the beam beta function, will give direct information on how sensitive is the PWFA process on the drive final focus positions, with the optimized parameters of this experiment. It would be helpful to give these spatial information together with the density map profile, which also plays a relevant role.

Due to experimental time constraints it was not possible to perform an accurate measurement of the waist location for each $k_x - k_y$ pair, and it is not possible to calculate the absolute waist location with sufficient accuracy from the read-back value of the quadrupole currents. We agree with the reviewer that, were such measurements available, this would be a valuable resource. Nevertheless one can make a simple estimate of the magnitude of this effect. Changing the focussing strength by 1% changes the focal length by approximately 1%. The final focussing quadrupole in x had a focal length of approx. 1.1 m, so a 1% change in k_x corresponded to a ~ 1.1 cm change in focal position. The final focussing quadrupole in y had a focal length of approx. 2.1 m, so a 1% change in k_y corresponded to a ~ 2.1 cm change in focal position. The following text was added to lines 196-199 of the manuscript: 'Simple calculations suggest that a 1% change in k_x, k_y corresponded to focal position shifts of approximately 1 and 2 cm respectively [c.f. the density profile in Fig. 1(c)].' Regarding changes in the beta function, please see our response to Reviewer 2 minor item 4.

2. Pg3, right column after "Given the extremely steep density ramps used in this work we suggest this as being close to the minimum peak current required to trigger DDRI in PWFAs". The referee definitively agrees with the analysis, but the sentence should be made more clear. It may help the introduction of the concept of minimum wakefield amplitude for sharp density downramps.

We and the reviewer agree that there is a clear minimum wakefield amplitude for injection into sharp density downramps, which naturally appears in the data as a minimum current for injection of charge. The sentence referred to by the reviewer has been removed and replaced on lines 219-226 with: "Electrons can only be trapped in the wakefield if their forwards velocity, which they obtain from their interaction with the high-current driver, exceeds the velocity of the back of the wakefield in the downramp. Given the extremely steep density ramps used, it is likely that 1.3 kA is close to the minimum driver current required in practice for DDRI".

3. Pg3, right column after "This was adjusted by changing the laser trigger timing in steps of 18.46 ns. $Q_i(\Delta t)$, shown in Fig. 2(d), increased from a few pC at $\Delta t = 10-20$ ps to a peak of 27 pC at $\Delta t = 18.46$ ns before dropping to approximately zero over the following 37 ns". As this is one of the key points of the density downramp scheme, the reader expects a sentence here about the time evolution of the electron density spike generated by the ionization laser pulse. At least Authors can add "(...as the timing is linked to the electron density spike profile, as shortly discussed in the next paragraph)"

We thank the reviewer for helping to clarify this point. The preceding sentence was updated to read ‘A final, powerful control over the injected bunch properties was the delay Δt between the arrival of the ionising laser beams and the driver bunch, since the timing determines the height and shape of the density spike, as discussed in the next section.’ on lines 233-237.

4. Pg4, left column, after “The resolution was $dz = 0.25 \mu\text{m}$, $dr = 0.4 \mu\text{m}$ with 2 azimuthal modes”. Authors should also include information about the box (cylinder) radius, length, the number of ppc ($N_r \times N_t \times N_z$) of the plasma and how they simulated the driver (as a distribution, with given macroparticles) and, most importantly, if they used a Lorentz boosted frame technique (in the case, as that technique also squeezes the plasma, they should also add information about the total number of macroparticles describing the accelerated beam, information which is relevant to understand the quality of the phase-space analysis)

The description of the simulations was updated on lines 268-276 to read:

“The resolution of the simulations was $dz = 0.25 \mu\text{m}$, $dr = 0.4 \mu\text{m}$ with 2 azimuthal modes. There were 2, 2 and 8 particles per cell in the r , z and θ directions, except in a small longitudinal region around the injection spike, where 4, 4 and 8 particles per cell were used. The box radius was $106.3 \mu\text{m}$ and its length was $354 \mu\text{m}$. A bi-Gaussian driver bunch was used with 304 pC charge, $I_{pk} = 1.9 \text{ kA}$, $\epsilon_n = 20 \text{ mm-mrad}$ and $\sigma_r = 16 \mu\text{m}$. It consisted of 300,000 macroparticles.”

The Lorentz boosted frame technique was not used. It is uncommon to use it in electron-beam-driven simulations, since the smallest longitudinal length scale of interest, the bunch length, is similar to the plasma wavelength, rather unlike the comparison between the laser and plasma wavelengths when considering laser-plasma accelerators. Furthermore, the smallest length scale of interest in these simulations was the injection spike in the plasma density profile, making the technique of Lorentz boosting the simulation particularly poorly suited to our case. Lines 257-261 were changed to simply reflect this. They now read ‘Imprecise knowledge of the initial conditions in this experiment precluded the use of magneto-hydrodynamics simulations, but insight in to how this general scenario affects injection was gained from a set of lab-frame quasi-3D FBPIC simulations.’

5. Pg5, right column. Related to the WarpX simulation, Authors should state the number of macroparticles which result in the accelerated beam (as for FB-PIC: did the use a Lorentz boosted technique?). Did Authors use impact ionization for the plasma?

A total of 134,049 macroparticles were in the accelerated beam and convergence tests focussed on the properties of the accelerated beam were performed. Lines 401-404 were updated to read ‘Figure 5(a) shows that the injected bunch, consisting of 1.3×10^5 macroparticles, was accelerated to a peak $E = 28.6 \text{ MeV}$, consistent with the experiment.’

For the reasons outlined in the answer to reviewer 1 minor change 4, the Lorentz boosted technique was not used. Lines 383-390 were updated to read: Figure 5 shows the longitudinal phase space, current profile and slice emittances of the injected bunch from the end of a lab-frame, fully 3D WarpX simulation, which used cell sizes $dx = dy = 0.39 \mu\text{m}$, $dz = 0.14 \mu\text{m}$, a box size of $200 \times 200 \times 400 \mu\text{m}$ and 1 and 4 particles per cell in the background plasma and plasma spike regions respectively.

Impact ionisation was not used. We did check if the electric field of the driver beam was large enough to cause significant ionisation, and found that this was unlikely to be the case due to the large emittance of the driver bunch (L. Boulton PhD Thesis, University of Strathclyde, DOI 10.48730/w6y5-9q35). We can exclude this as a possible injection mechanism since injected electrons were only observed when the transverse (plasma-spike-creating) laser was turned on. Furthermore, page 169-170 of the same thesis contains calculations of neutral gas scattering, which has a larger effect than electron-ion scattering for our parameters. The calculations showed that the normalised emittance growth from electron-neutral scattering was at the 30-40 nm level.

Reviewer 2

The manuscript by J. C. Wood et al., titled “Brightness-boosted electron bunches from a plasma accelerator” describes the first experimental demonstration that achieved both low emittance and energy spread.

As described in the manuscript, previous experimental demonstrations only achieved a large emittance (10 micron) and a few micron emittance with a large energy spread respectively. However, this demonstration achieved a 1 micron emittance with 30 pC charge. Also, the spectral density has improved by almost a factor of 10 compared to the previous researches. While these numbers are still much worse than what we can generate using conventional linacs, it is remarkable progress in this new scheme, which has big potential to overcome the limitations that conventional methods have. Furthermore, authors provided experimental study on system parameters that turned out to be critical for the total charge generated and explained the principle behind it using simulations. Such observations would significantly help subsequent researches on plasma-based electron sources. I believe the demonstrated results and analysis in this manuscript are indeed an important step towards the realization of plasma-based high-brightness electron sources, which are critical for future accelerators. Its impact would naturally be broad. Therefore, I believe this manuscript satisfies Nature Communications' publication criteria. However, there are two major issues and several minor items that must be addressed by authors.

We thank the reviewer for their positive comments and for their appreciation of the advances made in this work. We engage with the issues they raised below.

Major issue:

[1] The authors claim that the scheme they presented made a significant boost in brightness. However, this claim is not accurate. The drive beam used in the experiment had a charge of 300 pC, and its emittance was 20 micron. This emittance is almost 40 times larger than what I would expect from a 300 pC charge. If a nicely generated drive beam of the same charge has been prepared, the brightness of the drive beam could be improved by a factor of 1600. It is unfair to claim enhancement in brightness using such low-quality drive beam. Additionally, to claim that the scheme is really improving the brightness, such transverse emittance improvement on the drive beam must improve the main beam's up to some level. However, based on the description in this manuscript, I don't expect such improvements. Furthermore, the achievable brightness usually improves as the charge decreases. Thus, directly comparing the driver and injected bunches, having different charge level, is inappropriate. If we call such scheme as brightness boost, almost any schemes using two beams with difference charges can be considered as brightness boost. The wording "brightness boost" is incorrect and inappropriate. It must be eliminated for publication. The results of this manuscript are solid. I suggest that authors focus on their great demonstration of high-brightness plasma-based electron source.

We accept the comments about the phrase "brightness booster" and agree with the reviewer that we should concentrate on the explanation and implications of our results. Other reviewers had similar comments and we have partially dealt with this comment in our responses to them. We have changed the title and abstract of the manuscript to remove mentions of the brightness booster concept as discussed in our first minor response to reviewer 1. While reorganising the introduction as requested in the minor comments from reviewer 2 we similarly eliminated brightness booster references.

An isolated reference to the "brightness booster scheme" was found on lines 215-216 (of the original rather than the updated manuscript) and replaced by "this scheme", simply referring to DDRI.

References to brightness boosting were dropped in the main results sections entitled "Stability, Emittance and Brightness of the Injected Bunch". The brightness of the driver is simply stated as "In this experiment the driver $B_{3D,d} = 2.4 \text{ pC/MeV/mm-mrad}$ " on lines 377-378. Since the brightness booster narrative was dropped it no longer made sense to compare the brightnesses of the driver and injected bunches. Therefore figures 4(e), and 5(b) & (c) were removed. The rest of the figures were expanded to provide their details more clearly, and the figure captions were adjusted accordingly.

In the "6D Brightness Estimate from Particle-in-Cell Simulations" section comparisons to the driver brightness were removed.

In the "Discussion" section we removed the first half sentence that did read "Having provided the first experimental demonstration of the brightness booster concept". The remainder of this section until the final paragraph only focusses on improving the injected beam properties and future outlook. The final paragraph of the Discussion section has been mostly changed to simply restate the main results of

the paper, and now reads “In summary, we have presented experimental results from a beam-driven plasma accelerator in which a high-quality electron bunch was injected via density-downramp injection. The 1.3% energy spread, 14 pC MeV⁻¹ bunches were accelerated at approximately 1 GV m⁻¹ with high reproducibility. The mean normalised emittance of the injected bunch was measured to be (1.2 ± 0.1) mm-mrad. The combination of high charge per MeV and low emittance was a result of using a short ‘spike’ in the plasma density profile, with a peak density an order of magnitude greater than the surrounding plasma density. Particle-in-cell simulations showed that peak spike densities up to the drive beam density increased both the charge and brightness of the injected bunches.”.

[2] It was not clear to me what changes were made from other researches and what led to the improvements presented by authors. The authors need to provide more explanations regarding the changes they made compared to previous researches and how those changes contributed to achieving the enhancements.

The most important change was the use of density downramps with ratios of the peak plasma density to the following plasma density far greater than have been considered before. This is now stated in the final paragraph of the introduction (lines 113-118) as “This represents a significant step forwards in the quality of plasma-generated electron bunches. We achieved this using the DDRI mechanism with an especially high-density injection region with steep density gradients, far beyond what has been considered previously.” The mechanism behind this is explained in the section “Optimal Ramp Height & Length for DDRI”. For as long as the drive bunch density is greater than the plasma density in the downramp, injection can occur. The largest amount of charge is injected in the simulations (manuscript figure 3) when the peak downramp plasma density equals the beam density. This is also the mechanism behind the variation in injected charge with laser-to-electron-beam delay shown in figure 2(d) in the manuscript. This has been further summarised at the end of this section. Lines 324-328 have been altered to “The high charge per MeV injection resulting from these optimised density transitions, with high peak downramp densities equal to the bunch density, was a key factor in producing bright injected bunches in this experiment.”. Furthermore, we changed the final summary paragraph to more clearly state the main results, as detailed in the answer to your first major issue.

What is particularly unusual and interesting about the scan of ramp height and length shown in figure 3 is that the electron beam charge can be increased with no detriment to the emittance, which is shown via the close correlation between injected charge and brightness (compare figure 3(a) and (b)). This is in contrast to RF linacs where charge and emittance are intrinsically linked due to space charge forces in the injector, which are rapidly eliminated by multi-GV/m fields in a plasma wakefield. To further clarify our results, and to take into account this new insight which came about thanks to the reviewer’s question, the following sentences were added on lines 311-317: “Density downramps with peak densities $n_0 \sim n_b$ maximise both the injected charge and B_{3D} , as shown in Fig. 3(b), since the emittance varied little across this scan ($\epsilon_{n,x} = 0.4 - 0.5$ mm-mrad for $Q_i > 20$ pC). This is a surprising result, since in radiofrequency linac injectors the emittance of the bunch increases with charge, or at least current, due to space charge forces after its creation”.

Minor items:

1. The introduction doesn’t seem nicely organized. For example, the first two paragraphs almost repeat the same argument. The third one focuses on emittance and what can be possibly achieved, which is ok. However, the fourth paragraph goes back to why you need DDRI and beam-driven. Also, the fourth paragraph explains too much low level details without any other explanations. It was hard to read and hard to catch what authors wanted to say. I think such difficulty was coming from the paragraph and sentence structures. It would be great if the authors can improve this section.

The structure of the introduction has been changed. References to previous plasma acceleration work has been moved to the first paragraph to avoid repetition. Explicit discussion of driver-witness pairs has been omitted to help clarify what it is we studied in this work. The second paragraph discusses how the large accelerating fields of a plasma accelerator enable low-emittance bunch creation. The third explains density downramp injection, which was the chosen injection method in this work. More explanation was added in

this section as suggested in the third minor comment by this reviewer. The fourth paragraph provides a summary of the state of the field and the major results of this paper. We hope this structure improves the reading experience, and thank the reviewer for the suggestions. Due to the large changes made to the introduction, we did not highlight them individually this time.

2. “as they approach the back of the wake they experience a magnetic field from the radial current of sheath electron which act as a defocusing force”. Is it defocusing force not focussing force?

The force is defocussing, as explained in Ref. 23 of the manuscript (Xu et al., in particular their figure 3a). In that paper, following Lu et al. (references 33 and 34 from Xu’s paper), the transverse force on a plasma electron is described as consisting of 3 components: a defocussing force from the driver, a focussing force from the ions and a defocussing force proportional to the curvature of the wakefield potential (which Lu describes as being due to the magnetic field produced by the radial plasma electron current). At the back of the bubble the curvature of the wakefield potential becomes very large (commonly plotted as a rapidly varying longitudinal electric field), and only around this location does the defocussing force due to the radial plasma current exceed the focussing force of the ions. In a constant density plasma this causes most bubble sheath electrons to “bounce” from the axis rather than crossing it, which is observed in PIC simulations of plasma accelerators over a wide range of conditions, including with laser drivers. However in a density downramp, electrons that return to the axis with a large forwards velocity can be trapped in the wakefield while they are experiencing this defocussing force, which can lead to the injection of a low transverse momentum and thus low emittance bunch. A mathematical description is given by Xu which cannot be improved upon here.

3. It would be good to have more explanation of DDRI in the introduction. Currently it is forcing readers to find references to understand what that is, and it seems inappropriate for Nature Communications’ readers. It doesn’t have to be long but would be nice if you have what it is and why you do it (or what advantages are expected).

Please see our response to minor question 1 and the updated third paragraph of the manuscript, which now provides a basic overview of the DDRI process.

4. “the drive waist position in z while preserving its spot size”. If you change the focusing point in z direction, it changes the spot size. Do you mean by the change is ignorable? Or you actually made other compensation upstream? It would be better to describe it more precisely.

This is true, we wanted to imply that the change was small. This was achieved by having a large beta function in the final focussing quadrupoles, such that changing their focal length (strength) made a significant change to the focal plane position but only a small change to the beta function/ spot size at focus. Bunch tracking calculations in OCELOT show that a large change (in terms of the effect on injection) of 0.5% in k_x or 1% in k_y change the beta function at focus on the order of 1 mm, compared to a nominal beta function of 20 mm. Lines 188-193 of the manuscript were changed to read: ‘In Fig. 2(a) the final-focussing-quadrupole (FFQ) field strengths k in the x and y directions were varied. Due to the large β -functions in these quadrupoles changes in their strength mostly varied the driver waist position, while changing the β -functions at focus only at the 5% level.’

Reviewer 3

The article “Brightness- Boosted Electron Bunches from a Plasma Accelerator” describes an electron driven Plasma wakefield acceleration experiment using internal injection of the witness bunch. In injection mechanism is a density downramp injection. The main result is a 3D brightness measurement of the accelerated witness bunch which the authors claim is 4.8 times higher than the originally driving electron bunch. The paper is well written, rather clear and definitely original. The results presented are new and certainly important for the field of plasma acceleration. The paper describes a complicated experiment combined with a challenging data analysis. I therefore recommend the paper to be published. I would like the authors to consider nevertheless the following comments:

We thank the reviewer for their positive comments and their appreciation of the challenges involved in this experiment.

Introduction: To my opinion the authors describe somewhat inaccurate the limitation of conventional accelerators. For example a state of the art FEL is only a few 100 m long not several kilometers (for example Swiss FEL). Furthermore those conventional FEL's produce beams with an emittance of 100 nm for 100 pC charge. Therefore actually superior of what has been achieved in this paper. See for example: E. Prat, P. Dijkstal, M. Aiba, S. Bettoni, P. Craievich, E. Ferrari, R. Ischebeck, F. Löhl, A. Malyzhenkov, G. L. Orlandi, S. Reiche, and T. Schietinger Phys. Rev. Lett. 123, 234801, 2019. DOI: This facts should be mentioned which does not diminish the achievements of the publication. I find the term brightness-boost extremely misleading because there are two different beams, one loses energy and the other gains energy and is actually generated within the plasma. What is actually the brightness of the witness beam at its source if this can be simulated/determined.

We accept the reviewer's remarks about state-of-the-art FEL facilities and have made several changes to the manuscript. We have updated lines 28-30 to read 'As a result, the accelerators for free-electron lasers (FELs) are several hundreds of metres long, and particle colliders reach tens of kilometres in length.' Regarding the emittance of SwissFEL, we have updated the sentence now on lines 100-103 to read 'It has been shown in simulations that DDRI can produce bunches with $\epsilon_n < 100$ nm [refs], which is comparable to the very best conventional FEL driver linacs today [ref]', where the final reference is the one suggested by the reviewer above. Finally, we have made significant changes to the manuscript to mostly remove the term brightness booster- please see our responses to the first major issue from reviewer 2. Regarding the brightness of the witness beam at its source, please refer to our analysis of the evolution of the injected bunch's emittance and energy spread in simulations in our response to reviewer 1 major question 2.

Results: In this section I am missing the detailed data of the result. Which bunch charge, which emittance and energy spread was achieved. An image of both beams would be nice as well. It comes later but those are major results.

We thank the reviewer for this comment- we of course want our results to be as clear as possible. A summary of the results including the emittance, energy spread and spectral density of the injected bunches has now been included at the end of paragraph 4 of the introduction (it has not been highlighted due to the large changes that were made to the introduction). Unfortunately it is difficult to show images of the drive and injected beams that provide a fair, or simple-to-understand, comparison, since all images were taken with an imaging spectrometer, which imaged different energies in the two cases. Additionally, the dispersion changes as a function of energy. So, we find it best to rely on derived quantities rather than the images.

Optimal Ramp Height & Length for DDRI: This chapter is somewhat confusing. It seems to be a simulation study but it is not so clear how it is related to the experiment. Did the experiment had clear control on the ramp profile, do we know which profile gave the best result in the experiment. ? Please clarify this points.

This is an important point since we believe that the simulations in this section provide a strong scientific basis for understanding our experimental results. Similar comments to this one were made by the other reviewers, and we believe we have dealt with this reviewer's comment in our responses to the other reviewers. Please see our responses to reviewer 1 question 1, reviewer 1 minor question 4 and reviewer 4 question 1.

The experiment only had control over the ramp shape via the delay between the arrival of the ionising laser beams and the arrival of the electron beam. Please see the updated first paragraph of the Optimal Ramp Height & Length for DDRI which now contains a description of how these simulations relate to the experiment. We also refer back to our response to reviewer 1 minor question 3.

6D brightness estimate from PIC simulations: The authors like to suggest that the brightness boost in 6D could be more impressive than the measured one in 3D. The simulations seem to show quite asymmetric emittance values, wouldn't this suggest that the 3D measurement presented before might be optimistic because likely measured in the lucky plane?

We could only measure the emittance in the plane where simulations show the smallest emittance. This could be considered serendipitous, but it is also fair to say that improvements could be made to reduce the emittance in the other plane. It is also noteworthy that we have a lot of freedom to change parameters in the experiment, but we cannot optimise what we cannot measure.

The first figure in our response to reviewer 1 shows that the bunches are injected with similar emittance but in the y plane the emittance grows as it is poorly matched to the plasma, albeit with the caveat that we had no way to experimentally verify this. With more laser energy and adaptive optics the plasma spike itself could likely be shaped to reduce the emittance growth. We suspect that the reason for the mismatch is that the plasma spike is curved more strongly in y , where its width is of order the laser spot width, than in x , where its width is of order the laser Rayleigh range. So even using a larger focal spot size may decrease the emittance in the y plane.

Stability, Emittance and brightness of the injected bunch: It would be nice to make clearer what results are averaged over the data set and which are best shot results in this chapter. Is the 4.8 times boost a best shot result? What is the average of the data sample?

The 4.8 times boost (which is no longer reported, see our response to reviewer 2, although the brightness of the injected bunch is still reported) represents an average quantity, we do not report on best shot results unless we very explicitly state that we are doing so. The 3D brightness is built up from the spectral density of the injected bunch dQ_i/dE , the injected beam size after the plasma σ_x and its divergence $\sigma_{x'}$. We used the mean value of dQ_i/dE from 1000 measurements, the mean σ_x from 106 successful beam size measurements and the divergence inferred from an object plane scan consisting of 608 successful measurements. We feel that it would be unfair to estimate a best shot result, since not all of these quantities could be measured at the same time (although they were measured in the same conditions). It may not have been clear that this was the method we used in the manuscript, since we attempted to indicate that an averaging had been done using a bracket notation. Additional text has been added to the end of the "Stability, Emittance and Brightness of the Injected Bunch" section to convey this more clearly. In reference to the 3D brightness of the injected bunch, it reads: "Note that this value is constructed only from averaged values, with the exception of the injected bunch divergence, which was determined from a fit to a scan including ~ 600 data points."

Reviewer 4

The paper presents an experimental investigation of high-brightness electron bunches generation from a plasma accelerator, achieved through injection control of improved density transition using an additional injection laser to generate a higher density plasma spike. The injected electron bunch exhibits an improved brightness, with a 3D brightness 4.8 times greater compared to the driver electron bunch. The experimental results are elaborately discussed alongside particle-in-cell simulations. However, it is noted that a brightness-boosting scheme was previously proposed in [Phys. Rev. STAB. 7. 011301 (2004)] (Ref. [16] in the paper), and the experimental setup is almost identical to their previous work (Phys. Rev. Accel. Beams. 24. 101302 (2021)). As such, this paper appears to be more of an experimental verification work rather than one that extends the boundaries of knowledge. Therefore, it may not be suitable for publication in Nature Communications.

We thank the reviewer for engaging with the underlying physics of the experiment. It is true that previous publications have proposed using density-downramp injection, or other schemes such as Trojan Horse injection (Phys. Rev. Lett. 108, 035001, (2012)), to produce very bright electron bunches. We note that we provided this reference, plus the ones suggested by the reviewer here, as well as further similar references in the original manuscript. While the concept has been shown in theory and simulation, experimental confirmation was lacking that high spectral density (dQ/dE) bunches with low emittance can be internally injected within PWFAs. As in all physics fields, experimental proof is critical. Furthermore, the field of plasma accelerators often draws criticism due to the large shot-to-shot jitters in the accelerated beam parameters, which would limit their proposed applications. Therefore, it is not immediately clear from

a theoretical paper that the scheme can produce bunches with stable parameters in the presence of jitters. In addition to measuring bright bunches from this scheme, we also show that we can produce them very repeatably, which we think adds great value to the headline results.

While the setup is similar to previous work from the FLASHForward experiment by Knetsch et al. (Phys. Rev. Accel. Beams. 24. 101302 (2021)) the key innovation we made to create these bright injected bunches was to use extremely steep, high peak density downramps to inject them, which we explained with particle-in-cell simulations in the section titled 'Optimal Ramp Height & Length for DDRI'. We did not find this regime used or proposed elsewhere in literature. Furthermore, diagnostic upgrades were made since the experimentation of Knetsch et al.: the high resolution imaging spectrometer station (referred to as the HRSPEC in the manuscript) was installed to enable the measurement of small emittances. In Figure 2 of the manuscript and the surrounding text we studied how the system parameters affected injection. Three of these: drive bunch focal position, driver duration and the delay between the ionising laser and the arrival of the driver were not considered by Knetsch. We also highlight that Reviewer 2 positively commented on this aspect of the study and its implications for future work in this area.

There are several additional issues that need further discussion and revise:

1. In the scan simulations using the quasi-3D FBPIC code, which employs a cylindrical coordinate model, the simulated system needs to exhibit cylindrical symmetry. However, the spike plasma density created by the tightly focused injection laser deviates from cylindrical symmetry, which could impact the accuracy of the simulations.

Similar to our response to Question 1 from Reviewer 1, the goal of the q3D FBPIC simulations was not to simulate the experiment as accurately as reasonably possible. That was the goal of the full-3D WarpX simulation, the results of which were presented in Fig. 5. Instead we aimed to provide an explanation for the temporal variation of the injected charge, with the justified assumption that this variation affects the injection process by a changing downramp height and length. It was necessary to perform a 2D scan of these parameters since the ramp heights we used have not been reported in literature before and both ramp height and length impact the physics of injection, as detailed in the discussion of Fig. 3c. It was not possible to perform a 2D parameter scan of the more realistic WarpX simulations due to their prohibitive cost- the single simulation in Fig. 5 used approximately 1 month of our group's GPU time on the JUWELS Booster. Thus the reduced-dimensionality code was used. The density ramp in the FBPIC simulations was a linear ramp in z only, it did not vary as a function of r or angle. This has been clarified on lines 281-282 with the text: "The density varied only longitudinally, and not with radius or angle." We believe that the simplified FBPIC simulations also provide insight to other experiments and the wider field due to their conceptual simplicity. We hope that the additional text in the manuscript, detailed in our response to Question 1 from Reviewer 1, adequately describes our intentions for this simulation set.

2. The authors mention that, for the experimental results, although the emittance of the injected electron bunch is 17 times lower than that of the driver, the 3D brightness (B3D) only shows an increase of 4.8 times. This discrepancy is attributed to the deduced dQ_i/dE , which decreases as the energy spread increases. This challenge highlights the need to control the energy spread and phase space rotation of the injected electron bunch.

We agree with the reviewer that control of the energy spread is important. PWFAs generally do not feature phase space rotation. Since the driver and the witness propagate at very similar speeds, in the absence of significant driver evolution each slice of the witness stays locked at the same phase in the wakefield. In the absence of proper beam loading (Tzoufras et al., Phys. Rev. Lett. 101, 145002 (2008)) this causes the witness bunch to become chirped, although the slice energy spread remains constant. It has been found that an optimal length for low (projected) energy spread bunches from density downramp injection exists, which was calculated in publications such as Hue et al., Matter Radiat. Extremes 8, 024401 (2023). Over this length the energy spread is reduced as the initial chirp from the injection is cancelled out during acceleration, providing a potential solution to overcome the problem of improper beam loading. One may also speculate that good beam loading conditions may be found by tailoring the shape of the downramp to vary the current of the injected bunch along its length, although a study of this is understandably beyond

the scope of this manuscript. Therefore, schemes exist to produce a low projected energy spread bunch from density downramp injection after significant acceleration, in addition to a low slice energy spread. We also note that the energy spread of the accelerated bunch in the simplified simulation in figure 6 in the manuscript remains below 1% after acceleration to 635 MeV, over twenty times more energy gain than the measured bunches.

3. The authors suggest that their simulations indicate a potential >1000-fold increase in the 6D brightness (B6D) in their experimental configuration. However, it is noted that the emittance values used in the simulations are slice-normalized emittances, whereas the experimental results provide normalized transverse emittance values. This discrepancy could be misleading, as slice emittance is typically much lower than normalized transverse emittance. Furthermore, the simulations do not include the parameters of the driver, and it would be beneficial to compare the phase space distribution between the driver and the injected electron bunch.

Figure 5(a) plots the slice normalised emittance of the accelerated bunch. The projected emittances are also small as stated on lines 406-407. The 6D brightness from the simulation was calculated with these projected values although this was not stated in the manuscript. The manuscript was updated on lines 410-413 to read; 'The simulated $B_{3D} = 42 \text{ pC/MeV/mm-mrad}$, calculated using the projected emittances, was 3.7 times the experimental value, which used a conservative measurement of $\epsilon_{n,x}$.' The initial driver parameters are already stated on lines 397-401.

4. In Figure 6(b), the emittance appears different from that in Figure 5. It would be beneficial to include the brightness evolution in this figure for a more comprehensive analysis.

Figure 6 shows results from an idealised FBPIC (quasi-3D) simulation while Figure 5 shows results from a full 3D WarpX simulation that recreated the experiment as faithfully as possible. To make this more clear, lines 438-440 were updated to read 'The case II simulation from Fig. 3 was extended in FBPIC to a 195 mm long plasma using the same drive bunch.' To give a better picture of the evolution of the injected bunch brightness in this simulation we added the evolution of the relative energy spread to figure 6(b). Lines 448-452 were updated to read 'The FWHM energy spread of the bunch at $z = 195 \text{ mm}$ remains low at 0.6%, but is increased in absolute terms from a minimum value of 0.1 MeV shortly after injection to 3.7 MeV at the end of the plasma due to improper beam loading.' Given the approximately constant emittance this gives a full picture of the brightness evolution and its cause i.e the increasing energy spread. Please note that in this simulation no effort was made to optimise the energy spread of the injected bunch. Its low value demonstrates the potential of this scheme to produce high brightness bunches at higher energies.

5. The format of the references should be revised. For example, in reference [10], "Plasma Physics and Controlled Fusion" should be formatted as "...Plasma Phys. Control. Fusion..."

The references we used were produced by downloading the bibtex file from the paper URL wherever possible. When this was not possible our intention was to conform to the house style of Nature Communications. If inconsistencies with other journals are found, such as the one helpfully highlighted by the referee, then we would be happy to defer to the wishes of the editors.

Additional changes to the manuscript

We found two small mistakes in the analysis of figure 6, regarding the energy gain and energy spread. The electron energy reported in figure 6 and the surrounding text was too small by a factor of $m_e c = 0.511 \text{ MeV/c}$, which we have now corrected, and the energy spread of the injected bunch was overstated for both its minimum and maximum values. Specifically figure 6(a) was updated and the text on lines 445-452 was changed to 'The peak electron energy increases linearly with propagation distance to 635 MeV at $z = 195 \text{ mm}$ at a rate of 3.3 GeV m^{-1} . The FWHM energy spread of the bunch at $z = 195 \text{ mm}$ remains low at 0.6%, but is increased in absolute terms from a minimum value of 0.1 MeV shortly after injection to 3.7 MeV at the end of the plasma due to improper beamloading.'

Furthermore, a mistake was found on line 370 of the manuscript. Previously we stated that the resolution of the HRSPEC was $7\ \mu\text{m}$ but this did not account for the magnification of ~ 2.8 . The resolution is now correctly stated as $2.5\ \mu\text{m}$. The magnification was already correctly accounted for in the analysis of the bunch widths, however, and these have not changed. This does not change any conclusion, since the beam widths were much larger than the resolution so they were and are well resolved.

Yours faithfully, and on behalf of all of the authors

Jonathan Wood